# GNNShap: Scalable and Accurate GNN Explanations using Shapley Values

## ABSTRACT

Graph neural networks (GNNs) are popular machine learning models for graphs with many applications across scientific domains. However, GNNs are considered black box models, and it is challenging to understand how the model makes predictions. Game theory-based Shapley value approaches are popular explanation methods in other domains but are not well-studied for graphs. Some studies have proposed Shapley value-based GNN explanations, yet they have several limitations: they consider limited samples to approximate Shapley values; some mainly focus on small and large coalition sizes, and they are an order of magnitude slower than other explanation methods, making them inapplicable to even moderate-size graphs. In this work, we propose GNNShap, which provides explanations for edges since they provide more natural explanations for graphs and more fine-grained explanations. We overcome the limitations by sampling from all coalition sizes, parallelizing the sampling on GPUs, and speeding up model predictions by batching. GNNShap gives better fidelity scores and faster explanations than baselines on real-world datasets.

## CCS CONCEPTS

• **Computing methodologies → Neural networks**.

## KEYWORDS

GNN explainability, Shapley value, game theory

### ACM Reference Format:
Anonymous Author(s). 2018. GNNShap: Scalable and Accurate GNN Explanations using Shapley Values. In *Proceedings of Make sure to enter the correct conference title from your rights confirmation emai (Conference acronym 'XX).* ACM, New York, NY, USA, 13 pages. https://doi.org/XXXXXXX.XXXXXXX

## 1 INTRODUCTION

Graph Neural Networks (GNNs) are powerful models to learn representations of graph-structured data such as social [6, 21, 39], biological [7, 25], and chemical [4, 18, 41] networks. By capturing graph structures and node/edge features in an embedding space, GNNs achieved state-of-the-art performance on various tasks such as node classification, link prediction, graph classification, and recommendation [11, 14, 37, 40]. As with most deep learning models, a GNN represents a complex encoding function whose outputs cannot be easily explained by its inputs (graph structure and features).

As GNNs are widely used in scientific and business applications, understanding their predictions based on the input graph is necessary to gain users' trust in the model.

Like other branches of machine learning [16, 26, 31, 35], several effective GNN explanation methods have been developed in recent years, such as GNNExplainer [43], PGExplainer [17], PGM-Explainer [38], SubgraphX [46], and GraphSVX [5]. These methods often adapt explanation methods for structured data by incorporating graph topology information. For example, GraphLIME [13] is built on a popular linear model called LIME [26] and GraphSVX [5] is based on another popular method called SHAP [16].

Shapley's game-theoretic approach [32] is arguably the most widely-used explanation model where model predictions are explained by assuming that each feature is a "player" in a game where the prediction is the payout. Although Shapley value-based methods are known to provide good explanations, their main limitation is their computational costs. These methods require multiple perturbed input model predictions, which can be time-consuming. For deeper GNNs, the computational demand is even more prohibitive because of the rapid growth in the number of edges in the computational graph (commonly known as the *neighborhood explosion problem*[11]). To keep the running time reasonable, Shapley-based methods must use sampling, but sampling can also hurt the fidelity of the obtained explanations.

Considering this trade-off between fidelity and computational complexity, we develop GNNShap that is computationally fast and provides high-fidelity explanations for GNNs. GNNShap provides importance scores for all relevant edges when performing GNN prediction for a target node. GNNShap combines fast and effective sampling with batched model predictions to provide high-fidelity explanations for GNNs. We also employ parallel algorithms and pruning strategies to find explanations faster than other state-of-the-art (SOTA) methods.

The main contributions of the paper are as follows:

- We develop GNNShap, a Shapley-value based GNN explanation model that provides importance scores for all relevant edges for a target node.
- By improving the sampling coverage among all possible subgraphs, GNNShap improves the fidelity of explanations.
- GNNShap is two orders of magnitude faster than other Shapley-based explanation methods such as GraphSVX. This performance is obtained from our pruning strategies and parallel algorithms.
- GNNShap detects many unimportant edges that can be removed from the graph to expedite GNN inferences.

## 2 BACKGROUND AND RELATED WORK

Let $G(V, E)$ be a graph where $V$ is a set of nodes and $E$ is a set of edges with $|V| = N$. Let $A \in \mathbb{R}^{N \times N}$ be the sparse adjacency matrix of the graph where $A_{ij} = 1$ if $\{v_i, v_j\} \in E$, otherwise $A_{ij} = 0$.

Additionally, $X \in \mathbb{R}^{N \times d}$ denotes the node feature matrix. Without loss of generality, we consider node classification tasks where each node is mapped to one of $\mathbb{C}$ classes. If $f$ is a trained GNN model, the predicted class for a node $v$ is given by $\hat{y} = f(A, X, v)$.

## 2.1 Graph neural networks

GNNs use a message-passing scheme in which each layer $l$ has three main computations [3, 49, 50]. The first step propagates messages between the node pairs' $(v_i, v_j)$ previous layer representations $h_i^{l-1}$ and $h_j^{l-1}$ and relation $r_{ij}$ between the nodes $q_{ij}^l = \text{MSG}(h_i^{l-1}, h_j^{l-1}, r_{ij})$. The second step aggregates messages for each node $v_i$ from its neighbors $\mathcal{N}_{v_i}$: $Q_i^l = \text{AGG}(\{q_{ij}^l | v_j \in \mathcal{N}_{v_i}\})$. The final step of the GNN transforms the aggregated message and $v_i$'s previous representation $h_i^{l-1}$ via a non-linear transform function and updates the representation: $h_i^l = \text{UPD}(Q_i^l, h_i^{l-1})$.

## 2.2 Formulation of GNN Explanations

A computational graph $G_c(v)$ of node $v$ includes all information that a GNN model $f$ needs to predict $\hat{y}$ for $v$. For a two-layer GNN, a computational graph includes two-hop neighbor nodes and their node features. Formally, $G_c(v)$ computational graph with $A_c(v) \in \{0, 1\}^{a \times a}$ binary adjacency matrix, and $X_c(v) = \{x_j | v_j \in G_c(v)\}$ node features. A GNN explainer generates a small subgraph and subset of features $(G_S, X_S)$ for node $v_i$ for the prediction $\hat{y}$ as an explanation. We focus on node explanations in this work.

## 2.3 Shapley value and kernel SHAP

Shapley's game-theoretic approach [32] explains model predictions by assuming that each node, edge, feature is a "player" in a game where the prediction is the payout. A player's Shapley value can be computed using Eq. 1 by using weighted average of all possible marginal contributions of the player.

$$\phi_i = \sum_{S \subseteq \{1, \dots, n\} \setminus \{i\}}^{2^{n-1}} \frac{|S|!(n - |S| - 1)!}{n!} \left[ f(S \cup \{i\}) - f(S) \right] \quad (1)$$

Here, $n$ is the number of players, a coalition S is a subset of players, $|S|$ is the size of the coalition, and $f(S \cup \{i\}) - f(S)$ is the marginal contribution of player $i$'s to coalition $S$. The sum of the Shapley values equals the model prediction. The range of Shapley values is constrained by the model output. If the model output is a probability, then the value range will be between -1 and 1. The magnitude of Shapley values, except for their sign, indicates their importance for the model. Positive-scored players increase the model's output, while negative-scored players decrease the output. While Shapley value works well in explaining models, it needs to evaluate $2^{n-1}$ coalitions of players, which is infeasible when the number of players is large. Prior work addressed this computational challenge by approximating Shapley values using sampling. The most notable method is called kernel SHAP [16], which uses a surrogate linear model to approximate Shapley values.

Kernel SHAP is an additive method where the sum of the Shapley values gives the model prediction. The linear surrogate model $g$ is defined as:

$$f(x) = g(x) = \phi_0 + \sum_{i=1}^{n} \phi_i m_i, \quad (2)$$

where $m \in \{0, 1\}^{1 \times n}$ is a binary coalition mask that makes a coalition $S$, and $\phi$ is the surrogate model's parameters. The model parameters are the approximation of the Shapley values. $\phi_0 = f(\emptyset)$ is the case when there are no players. In addition, when a player missing ($m_i = 0$), the corresponding input of the model should be replaced with background data (e.g., expected value for the player). The linear model can be learned by minimizing squared loss in Eq. 3. Here, $\pi_{|S|}$ is called kernel weight and gives individual coalition weights for a coalition size. It gives more weight to small and large coalition sizes since it is easier to see the individual effect. Shapley values can be obtained by solving the weighted least squares problem [16].

$$\pi_{|S|} = \frac{n - 1}{\binom{n}{|S|} |S| (n - |S|)}$$

$$L(f, g, \pi_m) = \sum_{m \in M} \left[ f(S) - (g(S)) \right]^2 \pi_{|S|} \quad (3)$$

## 2.4 Related work

GNN explainability methods can be categorized into two main categories: instance-level and model-level explanations [45]. While instance-level explanations focus on an instance (e.g., a node explanation), model-level explanations, like XGNN [44], focus on the overall model's behavior. However, many studies focus on instance-level explanations. Instance-level explanation studies can be categorized into four classes:

Gradient/features works use gradients and/or features like model weights and attention scores as explanations. Saliency (SA) [2, 24], Guided Backpropagation [2], CAM [24], GradCAM [24], Integrated Gradients [36] are considered as gradient/features based explanations. The main limitations of gradients/features are gradients that can be saturated in some areas [34].

Decomposition methods LRP [2, 30], Excitation BP [24], GNN-LRP [29], and GCN-LRP [12] decompose the model prediction to the input layer using network weights. These methods need to access the model parameters, which makes them unsuitable for black-box models.

Perturbation methods, including GNNExplainer [43], PGExplainer [17], GraphMask [28], GrapSVX [5], SubgraphX [46], EdgeSHAPer [19], GraphSHAP [23], GStarX [47], FlowX [10], and Zorro [8] study prediction change when the input is perturbed. Surrogate methods Graph-Lime [13], RelEx [48], and PGM-Explainer [38] use a simple surrogate model to explain complex model predictions.

## 2.5 Shapley value-based GNN Methods

GRAPHSHAP [23] is a graph classification explainer that requires predefined motifs and assigns importance scores to motifs. However, mining the motifs is task-specific; it requires domain expertise.

EdgeSHAPer [19] is a graph explainer that considers edges as players. However, it's not based on Kernel SHAP; it computes each edge's Shapley value by computing marginal contributions using a certain number of samples. It needs to get two model predictions for each marginal contribution and repeat the process for each

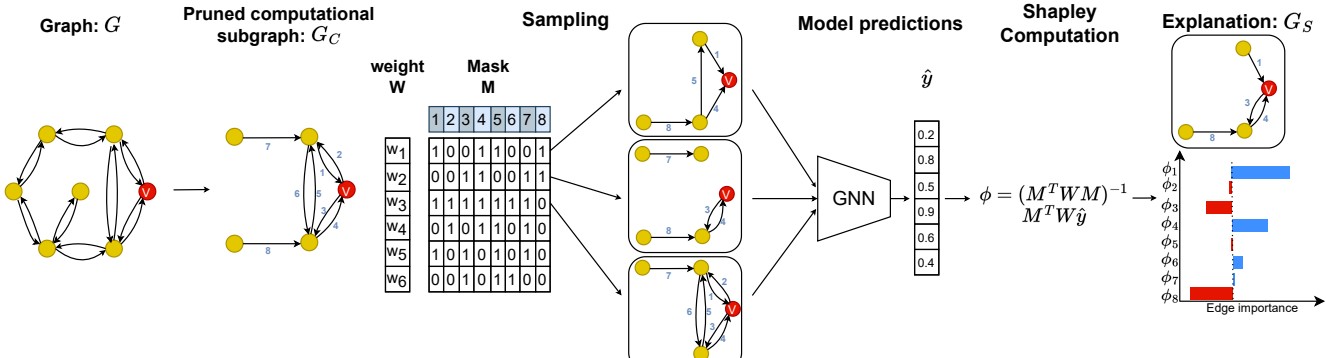

**Figure 1: GNNShap Overview: computing Shapley values for all edges in the computational graph for the target vertex $v$. We consider two-layer GNN explanations in the figure. The computational graph $G_c$ has eight directed edges (eight players). The mask matrix represents a sampling from all possible coalitions of players. Each sampling subgraph is used to get a prediction for $v$. The Shapley computation step then computes Shapley values based on GNN predictions for sampled coalitions.**

---

**Algorithm 1:** Overview of the GNNShap Algorithm

**Input:** $G = (A, X)$, $f$: GNN model, $n$: number of players, $p$ number of samples, $v_i$: the node to be explained, $l$: number of GNN layers, $b$: batch size.

**Output:** $\phi$: Shapley values for all players

1   $A_i, X_i \leftarrow PruneCompGraph(A, v_i, l)$    // find pruned computational graph

2   $p \leftarrow sum(A_i)$    // number of players (edges) in the $G_C$

3   $M, W \leftarrow Sample(n, p)$    // mask and sample weights

4   $\hat{y} \leftarrow GNN(M, A_i, X_i, b)$    // masked predictions

5   $\phi \leftarrow (M^T W M)^{-1} M^T W \hat{y}$    // weighted least squares

6   **return** $\phi$

---

Shapley value. Hence, it is computationally expensive, which makes it unsuitable for larger explanations.

GraphSVX [5] is another GNN explainer that can provide explanations for both nodes and node features. However, it mainly considers very small and very large coalitions. This can lead to sub-optimal solutions. Moreover, it requires much time to generate explanations, which makes it unsuitable for large graphs.

SubgraphX [46] targets to find the most important subgraph for the model using Shapley values. It uses a Monte Carlo tree search algorithm to explore subgraphs. However, SubgraphX is quite slow, even for middle-size graphs. Therefore it's not practical to use it for large graph explanations.

## 3 METHODS

### 3.1 Overview of GNNShap

Shapley value-based explanations for node $v$ in GNNs can be defined as follows: using the computational graph $G_c(v)$, the prediction $\hat{y}$ for $v$ is distributed among players, where players can be node features, neighbor nodes, and edges. Specifically, $\hat{y} = f(A_c(v), X_c(v)) = \sum_{i=0}^{n} \phi_i$, where $n$ is the number of players and $\phi_i$ is the Shapley value for player $i$. In this paper, we aim to identify

edges that are important for the prediction of the target vertex. In the edge-based explanations, edges in the computational graph are considered players in our explanation model. After computing Shapley values, we can obtain the explanation subgraph $G_S$ by selecting edges with higher Shapley values using top-k selection or using a threshold. Note that Shapley values can be positive or negative. Therefore, the absolute value of the Shapley scores should be used to determine importance.

Algorithm 1 shows an overview of GNNShap where we aim to explain the prediction of the target vertex $v_i$. Algorithm 1 shows four clear steps in GNNShap:

(1) **Obtaining pruned computational graph** (line 1 of Algorithm 1): for an $l$ layer GNN, we find the computational graph and prune redundant edges. This step is discussed in subsection 3.2.

(2) **Coalition sampling** (line 4): we sample subgraphs (coalitions) of the computational graph by increasing their coverage across all possible subgraphs. At this step, we create a $k \times n$ binary mask matrix $M$, where $k$ is the number of sampled subgraphs, $n$ is the number of players (edges) in the pruned computational graph, and $M[i, j]=1$ if the $j$th edge is present in the $i$th subgraph. The sampling phase also generates a weight vector $W$ where $W[i]$ stores the weight of the $i$th sample. This step is discussed in subsection 3.3.

(3) **Model prediction**: after samples are generated, we predict the class of the target node using each sample to generate the prediction vector $\hat{y}$ such that $\hat{y}[i]$ stores the prediction obtained using the $i$th sample. We using batching and parallelization to make this step faster. This step is discussed in subsection 3.4.

(4) **Shapley value computation**: we compute Shapley values for all edges in the computation graph using the following equation $\phi = (M^T W M)^{-1} M^T W \hat{y}$. This step is discussed in subsection 3.5.

Figure 1 shows an example of computing Shapley values using four steps discussed above.

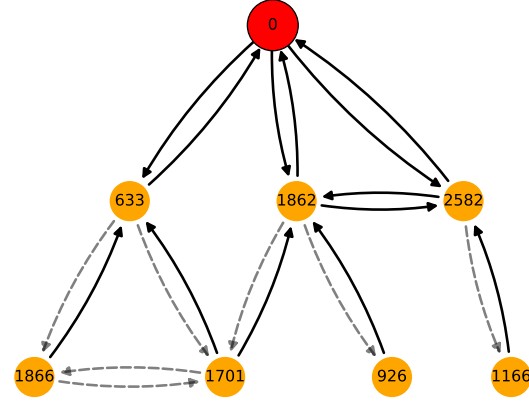

**Figure 2: Cora node 0's two-hop computational graph before and after pruning. The unpruned computational graph is created by keeping all nodes that are two hops away from the source node and all edges among those nodes. Dashed edges in the unpruned graph are redundant because their messages don't arrive at the source for two-layer GNNs. We prune the dashed edges in our computational graph.**

## 3.2 Pruning computational graphs

When explaining node $v$, all edges in $G_c(v)$ are considered players. Therefore, it is crucial to create $G_c(v)$ from the whole graph $G$ in a way that reduces computational complexity. Previous work [17, 43] considered all edges in the $l$-hop-induced subgraph as the computational graph, where $l$ is the number of layers in the GNN. However, such graphs may contain edges that do not carry a message to node $v$. We prune these redundant edges from the computational graph. Fig. 2 illustrates an example of a two-hop computational graph. While dashed edges are in the computational graph, their messages do not arrive at $v$. Hence, considering them as players will only increase computational complexity. In GNNShap, we prune these redundant edges, which can expedite the rest of the computations significantly. For example, the number of players (edges) reduces by 78 on average for a two-layer GNN on the Cora dataset.

## 3.3 Fast and efficient sampling for GNNShap

*3.3.1 Improving sampling coverage.* Sampling plays an important role in Shapley-based explanation methods since it is not possible to use all possible coalitions. For example, when using a graph with an average degree of $d$ on a 2-layer GNN, the computational graph of a vertex $v$ may have $O(d^2)$ edges (players), which results in $O(2^{d^2})$ possible coalitions.

Sampling used in previous Shapley-based GNN explanation methods such as GraphSVX only focuses on small and large coalition sizes by ignoring many coalitions that may contain useful information for explanations. GraphSVX includes samples from a mid-size coalition if a user-defined maximum coalition size is reached, yet

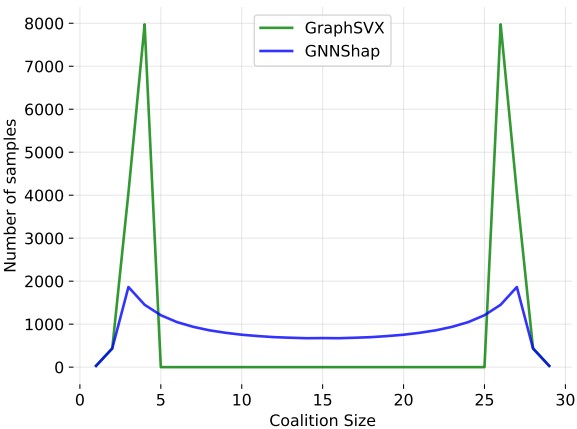

**Figure 3: Sample distribution figure for 30 players and 25,000 samples. While GNNShap distributes samples proportional to eq. 5, GraphSVX only samples from small and large coalition sizes.**

the number of samples has not been reached. These random samples are added by Bernoulli distribution without considering the coalition size.

We argue that an effective sampling for GNNs should sample from all possible coalitions because it can better capture the graph structure. To this end, we used ideas from the SHAP package [16] by distributing weights to all possible coalition sizes using eq. 4.

$$\rho_{|S|} = \frac{n-1}{|S|(n-|S|)}, \tag{4}$$

where $n$ is the number of players and $S$ is a coalition of players, and $\rho_{|S|}$ is the total weights for all samples of size $|S|$. This simple modification of Eq. 4 helps us sample from all possible coalition sizes. Next, our sampling approach based on kernel SHAP generates samples such that the number of samples is proportional to the total weight for the coalition size using eq. 5:

$$k_{|S|} = k * \frac{\rho_{|S|}}{\sum_{i=1}^{n-1} \rho_i}, \tag{5}$$

where $k$ is the total number of samples $k_{|S|}$ is the number of samples containing $|S|$ players. Fig.3 (the blue line) shows that this sampling approach indeed samples from all possible coalition sizes.

In our sampling approach, it is possible to generate more samples than the number of possible coalitions for very small and large coalition sizes. In this case, we redistribute surplus samples to the remaining coalition sizes. Finally, individual weights $w_{|S|}$ for samples are computed by distributing total coalition size weight to individual coalitions by 6.

$$w_{|S|} = \frac{\rho_{|S|}}{k_{|S|}} \tag{6}$$

This sample distribution strategy still gets more samples from small and large coalitions yet includes reasonable samples from the mid-sized coalitions. Even though individual coalitions with the same number of players get equal individual weight, mid-size coalitions

still contribute less in Shapley computation because of fewer samples taken from them.

We note that the original SHAP paper considered sampling without replacement to maintain unique samples. However, we did not observe a clear benefit when there are enough samples for GNN explanations. Hence, we use sampling with potential replacement to reduce the computational complexity of sampling. The output of sampling is a $k \times n$ binary mask matrix $M$, where $k$ is the number of sampled subgraphs, $n$ is the number of players (edges) in the pruned computational graph, and $M[i, j]$=1 if the $j$th edge is present in the $i$th subgraph. The sampling phase also generates a weight vector $W$ where $W[i]$ stores the weight of the $i$th sample.

*3.3.2 Fast sampling with parallelization.* We observe that sample generating is the slowest part of the Shapley-based explanations and requires more time when the number of samples and the number of players increase. Since we sample with replacement, each sample can be generated independently. To parallelize the sampling process on a GPU, we first distribute samples based on coalition sizes and generate all coalitions of a given size in parallel. We use lexicographical order algorithm [15, 20, 22] for fully sampled coalitions, which gives the ith combination without knowing the previous combination and random sampling for the other coalition sizes. We describe the sampling process in Algorithm 2 in the Appendix.

## 3.4 Fast model predictions using pruning and batching

*3.4.1 Prediction pruning.* At this step, we predict the class of the target node using each sample to generate the prediction vector $\hat{y}$ such that $\hat{y}[i]$ stores the prediction obtained using the $i$th sample. We observed that in some samples, the target node remains disconnected from the rest of the nodes when there is no incoming one-hop edge. These coalitions are still useful since the surrogate model learns that the marginal contribution of second-hop edges will be zero without a first-hop edge. However, it is not necessary to obtain the model predictions when the target node is disconnected, as they are equal to $f(\emptyset)$, which represents the model prediction without any neighbor information. In our model prediction, we prune this type of unnecessary model predictions, which reduces the number of required model predictions by 20%.

*3.4.2 Batching and parallel model predictions.* Shapley value-based approaches require model predictions for perturbed input, which is the most time-consuming step in the whole calculation. We expedite the model prediction step by batching samples and then running GNN predictions in a batch in parallel. To facilitate the batching, we create a larger block diagonal matrix by placing the adjacency matrices of the subgraphs within a batch along the diagonal. We also concatenate node features of all nodes in a batch. These enlarged feature matrix and block-diagonal adjacency matrix are used to predict classes of the target node with respect to a batch of samples. The main benefit of such batching is that it improves data locality and opportunities for parallel computations. We observed that batching made this step an order of magnitude faster than non-batched predictions. While batching makes GNN predictions faster, the creation of batches is itself an expensive process. To reduce the time to create batches, we start with the full l-hop-edges subgraph and then prune its edges using the mask matrix. This approach made the cost of batch creation insignificant when compared to the time needed for GNN predictions.

## 3.5 Efficient Shapley computations

The last step of the GNNShap is to compute $\phi = (M^T W M)^{-1} M^T W \hat{y}$. Note that $M$ is a $k \times n$ matrix with $k \gg n$ (that is, the number of samples is much larger than the number of players). Hence, the computational complexity of computing $M^T W M$ can be larger than computing the inverse of $M^T W M$ that is an $n \times n$ matrix. In our implementation, we stored $M$ as a dense matrix and performed the matrix multiplications in $M^T W M$ and $M^T W \hat{y}$ on the GPU. This made the multiplication part significantly faster than CPU-based multiplication. By contrast, we observed that the matrix inversion does not run fasted on the GPU when the number of players is large. This is because the current PyTorch implementation requires CPU synchronization for the inversion, which is costly $n$ is relatively large. To alleviate this problem, we train a weighted linear regression model on PyTorch instead of solving the equation when the number of players is over 5000. Note that the mask matrix $M$ is 50% sparse. However, our observations show that storing $M$ as a sparse matrix and performing sparse matrix multiplication is slower than dense matrix multiplications [9]. Hence, we opted to use dense computations in this step.

## 4 EXPERIMENTS

### 4.1 Datasets

We use six real-world datasets for the experiments. Cora, CiteSeer, and PubMed [42] are citation networks where nodes are papers, node features are bag-of-word representations of words in the paper, and edges are the citations of papers. We use the publicly available train, validation, and test splits. Coauthor-CS and Coauthor-Physics [33] are co-author graphs where nodes denote authors, edges denote coauthorships, and node features are keywords in the papers. We use 30 random nodes for each class for training and validation and the rest for testing. Facebook (FacebookPagePage) [27] is a verified page-page site graph. Nodes correspond to pages, edges are mutual likes, and node features are site descriptions. We use 30 random nodes for each class for training and validation and the rest for testing. Dataset statistics can be found in table 1. Since explanation generation for all baselines takes a lot of time, we only consider the first 100 test nodes for explanations.

### 4.2 Models

We use a two-layer GCN [14] with 16 hidden dimensions for Cora, CiteSeer, PubMed, and Facebook and 64 for Coauthor datasets. We apply ReLU as an activation function and 0.5 dropout in the training. We train the model for 200 epochs with a 0.01 learning rate. Model training and test accuracies are provided in Table 2. Since GNNShap views a GNN as a black box, it works seamlessly with other GNN models (see the appendix for its performance with GAT).

### 4.3 Evalution metrics

In this work, we utilize *Fidelity_−* (Eq. 7) and *Fidelity_+* (Eq. 8) metrics from [45] to evaluate the performance of our model. These

**Table 1: Dataset statistics. Players denote the number of edges for two-hop incoming edges for the first 100 test nodes.**

| Dataset | Nodes | Edges | Features | Classes | Avg players | Max players | Min players |
|---|---|---|---|---|---|---|---|
| Cora | 2708 | 10556 | 1433 | 7 | 159.08 | 298 | 5 |
| CiteSeer | 3327 | 9104 | 3703 | 6 | 25.17 | 262 | 2 |
| PubMed | 19717 | 88648 | 500 | 3 | 245.46 | 1106 | 4 |
| Coauthor-CS | 18333 | 163788 | 6805 | 15 | 161.61 | 1249 | 3 |
| Coauthor-Physics | 34493 | 495924 | 8415 | 5 | 428.61 | 10530 | 4 |
| Facebook | 22470 | 342004 | 128 | 4 | 858.34 | 7043 | 6 |

**Table 2: Model training and test accuracies**

| Dataset | Train | Test |
|---|---|---|
| Cora | 100.00 | 81.50 |
| CiteSeer | 99.17 | 71.00 |
| PubMed | 100.00 | 78.80 |
| Coauthor-CS | 94.44 | 90.22 |
| Coauthor-Physics | 100.00 | 95.33 |
| Facebook | 95.00 | 77.10 |

metrics measure the importance of edges in a graph, with $Fidelity_+$ focusing on important edges and $Fidelity_-$ focusing on less important edges. By removing important edges in $Fidelity_+$, we expect a significant change in the model prediction. Conversely, when dropping the least important edges in $Fidelity_-$, we expect only minimal changes in the prediction. Keeping top-k edges or thresholding approaches can be used to obtain the explanation $G_S$.

Fidelity scores can be computed for the ground-truth class or the predicted class. Since we explain the model's behavior for a prediction, we use the predicted class for the evaluation.

$$Fidelity_- = \frac{1}{N} \sum_{i=1}^{N} \left| f(G_C)_{\hat{y}_i} - f(G_S)_{\hat{y}_i} \right| \qquad (7)$$

$$Fidelity_+ = \frac{1}{N} \sum_{i=1}^{N} \left| f(G_C)_{\hat{y}_i} - f(G_{C \setminus S})_{\hat{y}_i} \right| \qquad (8)$$

### 4.4 Baselines

- Saliency (SA) [2, 24]: computes gradients with respect to node features and considers the sum of the gradients as node explanation.
- GNNExplainer [43]: uses mutual information to learn important edges and features. We train GNNExplainer for 200 epochs with a 0.01 learning rate for explanations.
- PGExplainer [17]: also uses mutual information and trains a neural network to provide explanations without requiring individual training. It provides edge explanations. We train PGExplainer for 20 epochs with a 0.05 learning rate on the training data.
- PGM-Explainer [38]: learns node importance by using a probabilistic graphical model. We use default settings for PGM-Explainer.
- GraphSVX [5]: a Shapley value-based GNN explainability method that jointly explains node and feature importance.

We use the "SmarterSeparate" algorithm with feature explanation disabled and set the maximum coalition size to three and the number of samples to 1000. Further increasing these numbers makes GraphSVX slower.
- SVXSampler is based on GraphSVX's "SmarterSeparate" in our framework. We use 10,000 samples with a maximum coalition size of three. It uses all our improvements except parallel sampling.

### 4.5 Test environment

We run all experiments on Ubuntu 18.04 with Intel(R) Core(TM) i9-7900X CPU @ 3.30GHz, 64 GB main memory, Nvidia Titan RTX 24GB (Driver Version: 460.39) and Cuda 11.7. We use Python 3.9.13, PyTorch 2.0.1, and PyTorch Geometric 2.3.1.

### 4.6 Evaluation protocol

The first 100 test nodes are used for explanations. Nodes with computational graph $G_C$ having less than two edges are excluded. Each experiment is repeated five times, and the average results are reported. A sparsity of 30% is used for $Fidelity_-$ scores since there are abundant unimportant edges. Conversely, the top 10 important edges are used for $Fidelity_+$ scores since critical edges for the prediction are scarce. Three different sample sizes (10,000, 25,000, and 50,000) are reported for GNNShap.

GNNShap batch size varies depending on the dataset. For Cora, CiteSeer, PubMed, and Facebook, the batch size is set to 1024. For Coauthor-CS, it is set to 512. Finally, for Coauthor-Physics, it is set to 128. The batch size had to be reduced for Coauthor datasets due to GPU memory limitations since the GNN model's hidden layer dimension is higher, and the number of maximum edges in $G_C$ is large. To compare our method for SA, PGM-Explainer, and GraphSVX baselines, we convert their node explanations to edge explanations by averaging two connecting node's scores as described in [1].

### 4.7 Fidelity Results

Table 3 presents the $Fidelity_-$ scores when 30% least important edges are removed from the graph. Since the explanation graph $G_S$ is obtained by dropping unimportant edges, smaller values are better in $Fidelity_-$ score. Table 3 shows that GNNShap outperforms all baselines for all three sample sizes by a significant margin. In most cases, using 10,000 samples leads to high-quality results. However, we observe a slight improvement in some cases when more samples are used. According to $Fidelity_-$ scores, the next best explanation model is SA or PGMExplainer. We observed that GraphSVX and

**Table 3:** $Fidelity_-$ scores for 30% sparsity (removing 30% least important edges): the smaller, the better. Emboldened numbers indicate the best performance while underlined numbers indicate second-best.

| Methods | Cora | CiteSeer | PubMed | Coauthor-CS | Coauthor-Physics | Facebook |
|---|---|---|---|---|---|---|
| SA | 0.021±0.000 | 0.037±0.000 | 0.030±0.000 | 0.046±0.000 | 0.017±0.000 | 0.036±0.000 |
| GNNExplainer | 0.039±0.001 | 0.105±0.002 | 0.071±0.002 | 0.104±0.001 | 0.020±0.000 | 0.062±0.001 |
| PGExplainer | 0.062±0.005 | 0.060±0.002 | 0.065±0.005 | 0.037±0.001 | 0.033±0.002 | 0.060±0.002 |
| PGM-Explainer | 0.025±0.001 | 0.038±0.002 | 0.029±0.002 | 0.030±0.003 | 0.014±0.002 | 0.035±0.005 |
| GraphSVX | 0.074±0.001 | 0.053±0.001 | 0.047±0.001 | 0.078±0.002 | 0.020±0.001 | 0.061±0.001 |
| SVXSampler | 0.062±0.000 | 0.045±0.001 | 0.093±0.000 | 0.097±0.001 | 0.040±0.000 | 0.130±0.001 |
| GNNShap 10k | 0.009±0.000 | **0.020±0.000** | 0.011±0.000 | 0.015±0.000 | **0.005±0.000** | 0.015±0.000 |
| GNNShap 25k | 0.009±0.000 | 0.022±0.000 | **0.010±0.000** | **0.013±0.000** | 0.005±0.000 | 0.015±0.000 |
| GNNShap 50k | **0.008±0.000** | 0.020±0.000 | 0.011±0.000 | 0.015±0.000 | 0.005±0.000 | **0.013±0.000** |

**Table 4:** $Fidelity_+$ scores for identifying top10 important edges: the higher, the better. Emboldened numbers indicate the best performance while underlined numbers indicate second-best.

| Methods | Cora | CiteSeer | PubMed | Coauthor-CS | Coauthor-Physics | Facebook |
|---|---|---|---|---|---|---|
| SA | 0.108±0.000 | 0.128±0.001 | 0.086±0.000 | 0.123±0.000 | 0.057±0.000 | 0.062±0.000 |
| GNNExplainer | 0.036±0.002 | 0.111±0.002 | 0.047±0.000 | 0.053±0.000 | 0.024±0.000 | 0.039±0.001 |
| PGExplainer | 0.081±0.005 | 0.112±0.003 | 0.056±0.003 | 0.128±0.003 | 0.036±0.002 | 0.054±0.005 |
| PGM-Explainer | 0.133±0.013 | 0.134±0.007 | 0.073±0.007 | 0.141±0.011 | 0.059±0.003 | 0.065±0.004 |
| GraphSVX | 0.178±0.000 | 0.159±0.000 | **0.138±0.001** | 0.189±0.001 | 0.059±0.000 | 0.120±0.002 |
| SVXSampler | 0.200±0.001 | 0.167±0.000 | 0.131±0.000 | 0.218±0.001 | 0.097±0.000 | 0.168±0.000 |
| GNNShap 10k | **0.206±0.000** | 0.167±0.000 | 0.136±0.000 | 0.228±0.000 | 0.102±0.000 | **0.175±0.000** |
| GNNShap 25k | 0.204±0.000 | 0.167±0.000 | 0.134±0.000 | 0.227±0.000 | **0.103±0.000** | 0.173±0.000 |
| GnnShap 50k | 0.206±0.000 | **0.168±0.000** | 0.136±0.000 | **0.229±0.000** | 0.103±0.000 | 0.175±0.000 |

**Table 5:** Total explanation times in seconds for the first 100 test nodes. PGExplainer training time is provided in parenthesis.

| Methods | Cora | CiteSeer | PubMed | Coauthor-CS | Coauthor-Physics | Facebook |
|---|---|---|---|---|---|---|
| Saliency | 0.35±0.01 | 0.33±0.01 | 0.35±0.00 | 0.39±0.00 | 0.61±0.01 | 0.33±0.00 |
| GNNExplainer | 95.95±0.29 | 96.52±0.09 | 97.39±0.62 | 191.08±0.16 | 386.45±0.24 | 105.81±0.37 |
| PGExplainer | 0.40±0.00 (22.50) | 0.51±0.00 (34.63) | 1.55±0.01 (58.00) | 6.80±0.30 (1832.40) | 16.79±0.02 (1607.05) | 0.47±0.00 (25.87) |
| PGM-Explainer | 733.69±0.88 | 1177.79±0.98 | 4793.42±3.59 | 8118.04±23.68 | 16958.51±26.93 | 5539.50±3.18 |
| GraphSVX | 908.45±0.61 | 259.65±0.98 | 1282.08±1.27 | 1668.58±0.65 | 4056.97±1.33 | 3381.89±3.25 |
| SVXSampler | 24.11±0.05 | 12.07±0.13 | 26.31±0.09 | 32.29±0.09 | 68.19±0.19 | 100.43±0.30 |
| GNNShap 10k | 6.68±0.08 | 3.61±0.11 | 5.24±0.09 | 19.18±0.09 | 46.18±0.29 | 15.34±0.11 |
| GNNShap 25k | 12.65±0.11 | 5.52±0.23 | 9.09±0.18 | 45.16±0.09 | 112.58±0.28 | 29.26±0.07 |
| GNNShap 50k | 22.59±0.08 | 8.37±0.11 | 15.65±0.37 | 88.50±0.11 | 223.18±0.25 | 52.33±0.04 |

SVXSampler used in GNNShap perform poorly. The results show that GNNShap is very effective in identifying unimportant edges in a GNN explanation. We further validate the superiority of GNNShap by showing the $Fidelity_-$ scores at various sparsity levels in Fig. 4 for the Cora dataset. We observe that GNNShap outperforms all baselines at all sparsity levels. We observe similar results for all other datasets in the appendix.

Table 4 presents the $Fidelity_+$ scores. In $Fidelity_+$ scores, higher prediction change is expected since we drop the ten most important edges. According to the $Fidelity_+$ scores, GNNShap outperforms all baselines for five out of six datasets, except for PubMed, where

GraphSVX slightly outperforms GNNShap. Fig. 11 in the Appendix shows detailed $Fidelity_+$ results at various top-k level. We observe that GNNShap is one of the best performers in identifying the most important edges for GNN explanations.

Overall, Shapley-based methods GraphSVX, SVXSampler and GNNShap preform much better than their competitors when identifying important edges for GNN predictions. GraphSVX is the best method among the baselines for $Fidelity_+$ scores, while it under performs for $Fidelity_-$. Therefore, we can conclude that Shapley-based approaches tend to lead to better results in finding the most

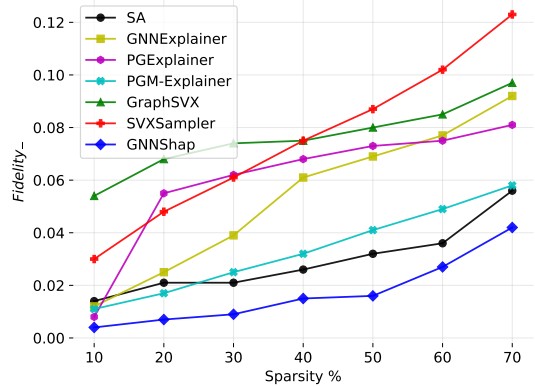

Figure 4: $Fidelity_-$ scores on Cora dataset for sparsities. The lower result is desired for $Fidelity_-$. GNNShap gives the best result for all sparsity levels.

important edges. However, GNNShap's sample distribution leads to better results in general.

## 4.8 Explanation Times

Table 5 shows the total explanation times for the first 100 test nodes. GNNShap is significantly faster than GNNExplainer, PGMExplainer, and GraphSVX. Our parallel sampling and pruning strategies reduce the explanation times drastically. Although Shapley-based approaches are generally considered slow, GNNShap is still faster even when using 50,000 samples. Most importantly, GNNShap is up to 100× faster than GraphSVX that is the overall second best performer according to $Fidelity_-$ and $Fidelity_+$ scores. SVXSampler is implemented inside GNNShap's framework. Hence, it runs much faster than GraphSVX.

SA is computationally efficient since it only requires a forward pass. Similarly, PGExplainer is computationally efficient because it can provide global explanations without individual learning after training. However, GNNShap Fidelity results are significantly better than those of PGExplainer and SA.

## 4.9 Improving prediction confidence based on edge importance

As shown in Fig. 4, GNNShap is very effective in identifying unimportant edges. We expect that prediction probabilities should increase when the negative contributed edges are removed from the graph. Fig. 5 confirmed this hypothesis where removing unimportant edges improve the prediction confidence for all nodes. Thus, GNNShap can help us sparsify the graph, which helps reduce the computational complexity of GNN inference while improving prediction confidences.

## 4.10 Explanation Visualization

GNNShap is able to visualize explanations. Fig. 6 shows an explanation for Cora node 37. While blue edges reduce the prediction probability, red edges reduce the probability. The visualization can help to identify undesirable outcomes of the model.

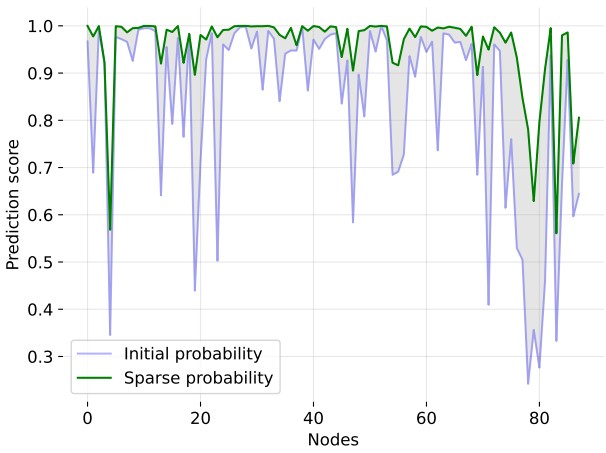

Figure 5: Cora model prediction probability improvement for nodes when edges with negative Shapley value are dropped.

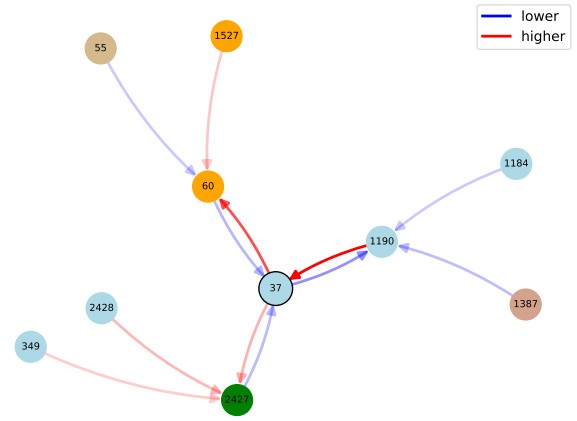

Figure 6: Explanation Graph Visualization for node 37. Node colors show classes. While blue edges reduce the prediction probability, red edges reduce the probability.

## 5 CONCLUSION

Shapley-value based explanations have been very successful in almost all branches of machine learning. However, their use was limited in GNN explanations because of their high computational costs and difficulties in finding unimportant edges. This paper presents GNNShap that addresses both problems by first using an effective sampling strategy and then developing faster algorithms using pruning and parallel computing. Through a comprehensive evaluation, we demonstrate that GNNShap achieves state-of-the-art performance on various node classification tasks quickly and accurately.

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

# A  COMPUTATIONAL GRAPH PRUNING

Table 6 shows the reduction of players for the two-hop computational graph when pruning is applied. Our pruning strategy only selects edges that their message reaches to the node in l-hops. The pruning reduces the number of players on average by over 50%. The reduction increases in larger graphs such as Coauthor-Physics and Facebook.

# B  MODEL PREDICTION

## B.1  The Impact of Prediction Pruning

Table 7 presents the percentage of pruned predictions. The table indicates that for a significant number of samples, we do not need

**Table 6: Average number of players (edges) reduction for two-hop computational graphs on the test nodes of datasets.**

| Dataset | Cora | CiteSeer | PubMed | Coauthor-CS | Coauthor-Physics | Facebook |
|---|---|---|---|---|---|---|
| before pruning | 124.11 | 52.4 | 262.66 | 715.71 | 2333.51 | 7373.78 |
| after pruning | 45.73 | 21.59 | 81.71 | 162.96 | 365.34 | 948.33 |
| reduction | 78.38 | 30.81 | 180.95 | 552.75 | 1968.17 | 6425.45 |
| **reduction %** | **63.15** | **58.80** | **68.89** | **77.23** | **84.34** | **87.14** |

to obtain the model predictions of over 20% of the samples to obtain the prediction.

## B.2 Sequential vs Batch Model Prediction

To evaluate the effect of batching on coalition predictions, we run coalition predictions in sequential and batched. Since sequential inference takes a lot of time, we consider 10,000 samples. Table 8 shows that we see over 100 times speed-up for some datasets. Due to GPU memory, we had to reduce the batch sizes for Coauthor-Physics (128) and Coauthor-CS (512). With a GPU with more memory, the speed-up can be further improved for these datasets.

## C SAMPLING

### C.1 Parallel Sampling

Our parallel sampling algorithm is presented in Algorithm 2 and 3. In the first step, Algorithm 2, samples are distributed to coalition sizes, and a cumulative sum of samples of coalition sizes is kept in a vector. Then, the GPU kernel code is called to start the parallel sampling; Algorithm 2. Each GPU thread first computes its chunk range. If the range requires samples from the to-be-fully-sampled coalition size, it creates the sample using LexicographicOrder. For random sample cases, it only needs to know the coalition size |S| and creates a random sample by setting |S| values to true. When a sample is added to the mask matrix, its complementary sample is also added to achieve symmetric sampling. Therefore, no computation is needed for half of the samples.

### C.2 GNNShap Explanation Time Breakdown

Fig. 7, illustrates an explanation time breakdown for 307 players. The figure shows that the most time-consuming part of a GNNShap explanation is sequential sampling. However, after parallelizing, the sampling becomes the least time-consuming operation for an explanation.

### C.3 Effect of Unique Samples

SHAP [16] ensures that each sampled coalition is unique. Although the uniqueness of coalitions is a desirable property, verifying unique coalitions increases computation time. Fig. 9 shows that checking the uniqueness increases computational time, while there is no clear benefit on fidelity scores when there is a reasonable number of samples. Furthermore, controlling the uniqueness of coalitions during sampling makes parallelizing the sampling quite difficult. Therefore, we do not control the uniqueness of coalitions during sampling.

---

**Algorithm 2:** GNNSHAP Sampler Algorithm

**Input:** $n$: number of players, $p$: number of samples.
**Output:** $M$: boolean mask matrix, $W$: weight vector

1  $bins \leftarrow p * eq.5$       // distribute samples to coalition sizes using eq. 5
2  $r \leftarrow 0$                     // random sampling start index
3  $coalSizeInds \leftarrow []$     // coalition size sample start indices
4  **for** $c \leftarrow 1 \rightarrow n/2$ **do**
5    **if** *more samples than possible coalitions for c* **then**
6      redistribute extra samples to the remaining bins
7      $coalSizeInds[c] \leftarrow r$
8      $r \leftarrow r + \binom{n}{c}$
9      $W[coalSizeInds[c] : r] \leftarrow eq.3$
10   **else**
11     $coalSizeInds[c] \leftarrow coalSizeInds + bins[c]$
12  $W[r : p/2] \leftarrow (0.5 - sum(W[: r]))/(p/2 - r)$  // distribute remaining weights to random samples
13  $W[p/2 :] \leftarrow W[r : p/2]$          // symmetric samples' weights
14  GPUSample(M, coalSizeInds, randInd, n, p)
15  **return** $M, W$

---

**Algorithm 3:** GPUSample Algorithm

**Input:** $n$: number of players, $p$: number of samples, coalSizeInds: coalition start indices, randInd: random sampling start index
**Output:** $M$: boolean mask matrix, $W$: kernel weight vector

1  $s \leftarrow$ compute chunk start index
2  $e \leftarrow$ compute chunk end index
3  $i \leftarrow s$
4  **for** $i \rightarrow randInd$ **do**                    // fully sampled coalitions
5    $c \leftarrow$ compute coalition size using coalSizeInds
6    M[i] = LexicographicOrder(n, i - coalSizeInds[c])
7  **for** $i \rightarrow e$ **do**                          // random sampling
8    $c \leftarrow$ compute coalition size using coalSizeInds
9    M[i, rand(c)] = true                          // set random c index as true
10 **return** $M, W$

## D EXPLANATIONS WITH OTHER GNN TYPES

To demonstrate that GNNShap can generate fast explanations for other GNN types, we use a two-layer GAT model with 16 hidden

**Table 7: Average percentage of pruned predictions of 100 node explanations for 25,000 samples. If a target node is disconnected in a coalition, there is no need to get the prediction since they are equal to $f(\emptyset)$.**

| Dataset | Cora | CiteSeer | PubMed | Coauthor-CS | Coauthor-Physics | Facebook |
|---|---|---|---|---|---|---|
| Pruned prediction % | 28.89 ± 0.00 | 23.44 ± 0.00 | 22.97 ± 0.00 | 23.75 ± 0.00 | 23.71 ± 0.00 | 27.31 ± 0.00 |

**Table 8: Sequential versus batched total model prediction times of 100 explanations in seconds for 10,000 samples**

| Methods | Cora | CiteSeer | PubMed | Coauthor-CS | Coauthor-Physics | Facebook |
|---|---|---|---|---|---|---|
| Sequential | 519.29 | 366.29 | 543.35 | 518.58 | 554.45 | 520.5 |
| Batched | 6.68 | 3.61 | 5.24 | 19.18 | 46.18 | 15.34 |
| **Speedup** | 77.74x | 101.47x | 103.69x | 27.04x | 12.01x | 33.93x |

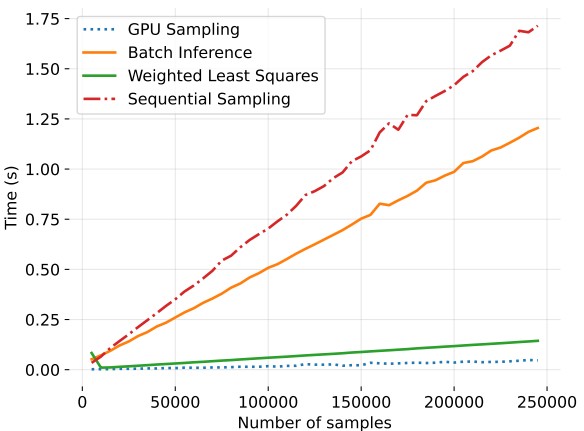

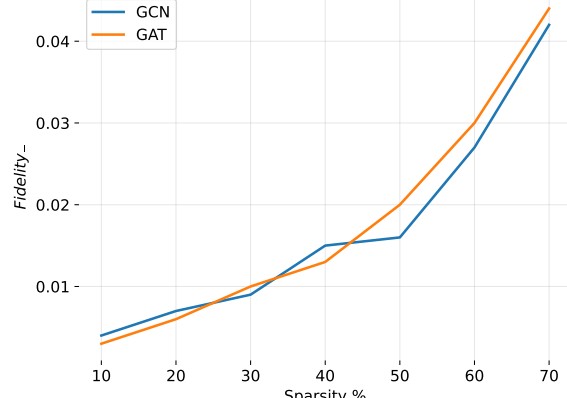

**Figure 7: Explanation Time breakdown of GNNShap computation for the increasing number of samples. Explanation computational graph pruning is not included since its time is negligible. The timing breakdown reveals that the most time-consuming part was the sampling process. However, after parallelization, the sampling process became negligible.**

**Table 9: Total annotation generation time of GNNShap for GCN and GAT on 100 Cora dataset nodes.**

| GNN | GCN | GAT |
|---|---|---|
| GNNSHAP 10k | 6.68±0.08 | 9.09±0.09 |

layers of multi-head-attentions with 8 multi-head-attentions for the Cora dataset. We train the GAT model for 200 epochs with a 0.005 learning rate. While the GAT model training accuracy is 100%, the test accuracy is 81.4%. GCN model details can be found in section 4.2, and accuracy is in Table 2. The batch size of GNNShap is set to 1024 for the experiment. Table 9 provides GNNShap computation times for 10,000 samples. GNNShap generates explanations faster for GCN because the GAT model requires more computation than GCN. Yet, GNNShap generates 100 explanations on the Cora dataset in under 10 seconds. Quality-wise, GNNShap gets similar $Fidelity_-$ scores for both models, as can be seen in Figure $Fidelity_-$. This result shows that for similar accuracy models, GNNShap is able to distinguish important and less important edges successfully.

**Figure 8: GNNShap 10,000 samples $Fidelity_-$ scores for GCN and GAT model. GNNShap gets similar fidelity scores for both methods.**

## E  FIDELITY SCORES

Fig. 10 shows $Fidelity_-$ scores for all datasets. It is obvious that GNNShap outperforms the baselines. However, GraphSVX and SVXSampler used in GNNShap perform poorly. The results show that our sampling strategy is very effective in identifying unimportant edges.

Fig. 11 shows $Fidelity_+$ scores for multiple top-k levels. The results indicate that GNNShap is one of the best performers for $Fidelity_+$, except for PubMed, where GraphSVX is slightly better. Overall, Shapley value-based approaches are good at identifying important edges. However, GNNShap performs well on both $Fidelity_-$ and $Fidelity_+$.

## F  REPRODUCIBILITY

We provide the necessary parameters to reproduce our experiments for the model in Section 4.2, baselines, and their parameters in Section 4.4. We also provide GNNShap-specific parameters in Section 4.6.

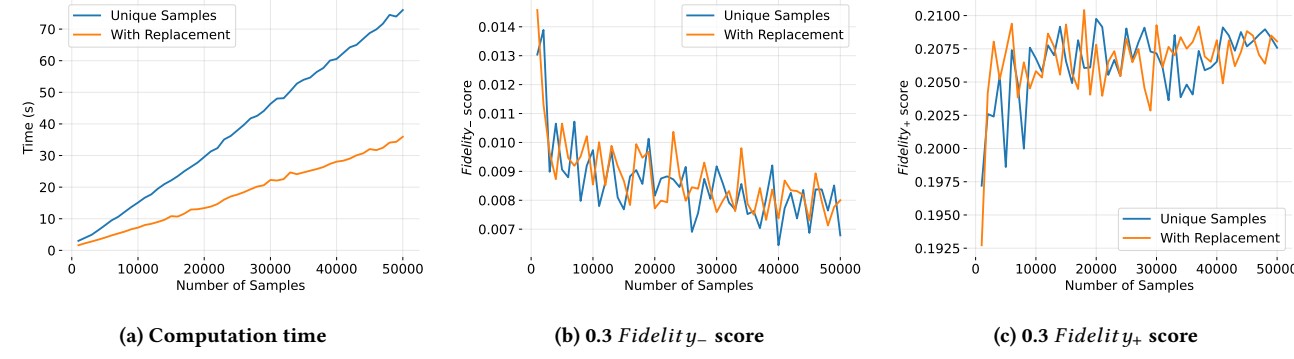

(a) Computation time

(b) 0.3 $Fidelity_-$ score

(c) 0.3 $Fidelity_+$ score

**Figure 9: Unique coalitions vs. samples with replacement effect on Cora dataset for various number of samples. (a) shows computation time of sampling when uniqueness is checked, (b) shows $Fidelity_-$ score for 30% sparsity, (c) shows $Fidelity_+$ score for top 10.**

**Figure 10: $Fidelity_-$ scores for sparsities. The lower result is desired for $Fidelity_-$. GNNShap gives the best result for all sparsity levels.**

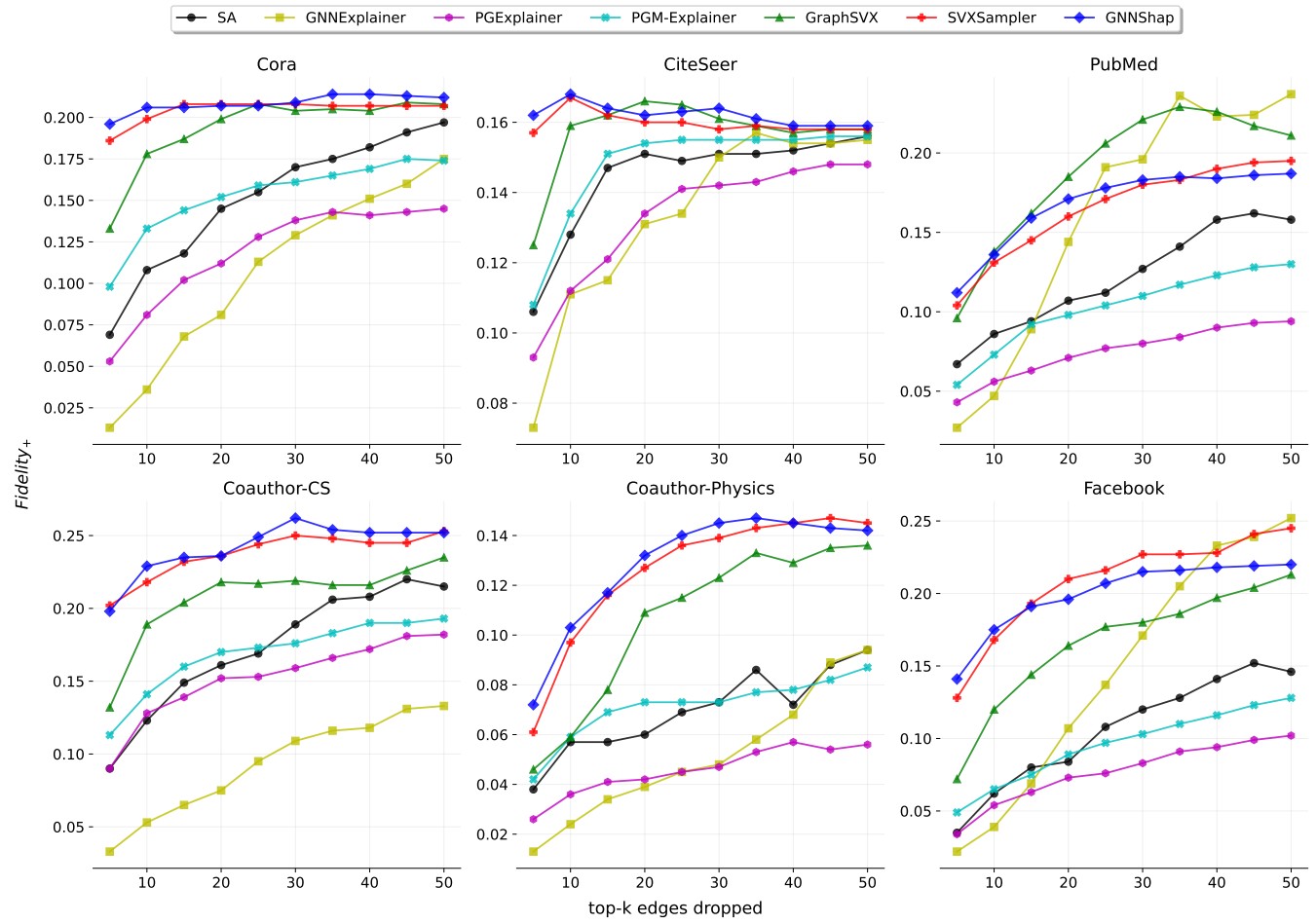

**Figure 11:** $Fidelity_+$ **scores for different top-k levels. The higher result is desired for** $Fidelity_+$**. GNNShap gives the best result for all sparsity levels.**

