# OpenReview forum: "GNNShap: Scalable and Accurate GNN Explanation using Shapley Values"
_ACM.org/TheWebConf/2024/Conference — TheWebConf24_

### Official Review · Reviewer_DyPd · 2023-11-22

**Novelty:** 3
**Technical Quality:** 4

**Review:**

Graph neural networks (GNNs) are powerful to make predictions over graphs. This work introduces GNNShap, an approach that provides explanations for GNNs. GNNShap employs comprehensive sampling across all coalition sizes, parallelizes sampling on GPUs, and enhances prediction speed through batching.

Pros:

- The paper is easy to read.

- The method is faster than the previous Shapley Value based method for GNN explanation.

- The efficiency is achieved through using only computation graphs.

Cons:

- It is hard to evaluate the novelty of the proposed method.

- The method is motivated via efficiency, but the datasets used in the experiments are small.

- Some of the experimental design needs more explanation. The results overall are also not strong.

**Questions:**

1. Could you please explain how the running times are being reported for all the algorithms? Usually, the learning-based algorithms are much faster. The numbers don't make sense.


2. Could you please elaborate the exact differences with GraphSVX given that Shapley value has been used in the past for this problem as well as the sampling techniques are know? There are some papers (e.g., [1]) that even prove guarantees for sampling to compute Shapley value fast.

3. Do you have more results for Fidelity? The experiments only involve 30% removal. What happens in other settings?

[1] Mitchell, Rory, Joshua Cooper, Eibe Frank, and Geoffrey Holmes. "Sampling permutations for shapley value estimation." The Journal of Machine Learning Research 23, no. 1 (2022): 2082-2127.

**Reviewer Confidence:**

4: The reviewer is certain that the evaluation is correct and very familiar with the relevant literature

**Scope:**

3: The work is somewhat relevant to the Web and to the track, and is of narrow interest to a sub-community

---

### Official Review · Reviewer_YZ7F · 2023-11-23

**Novelty:** 5
**Technical Quality:** 5

**Review:**

Considering the overwhelming time consumption of existing Shapley value-based GNN explanation methods, this paper introduces several technics to reduce the time consumption. The proposed technics are well motivated and reasonable. But the technical contribution seems to be limited.

**Pros**:
1. GNN explanation is a critical problem in Graph learning. The proposed technics in this paper are well motivated and reasonable.
2. Authors propose many mechanisms to reduce the time consumption of Shapley value-based GNN explanation method.
3. Authors conduct many experiments to evaluate the proposed method.


**Cons**:
1. Although the proposed mechanisms to reduce time consumption are technically reasonable, they are either simple and intuitive or just simple modification of existing methods. The technical contribution of the proposed mechanisms are kind of limited.
2. Lacking ablation study. The effects of different components in the proposed method are unclear.
3. The time consumption of GNNShap on several datasets is greater than that of SVXSampler.

**Questions:**

As listed in Cons.

**Reviewer Confidence:**

3: The reviewer is confident but not certain that the evaluation is correct

**Scope:**

3: The work is somewhat relevant to the Web and to the track, and is of narrow interest to a sub-community

---

### Official Review · Reviewer_eQAE · 2023-11-23

**Novelty:** 4
**Technical Quality:** 5

**Review:**

This paper proposes a Shapley Value-based GNN explanation method. The authors introduce several mechanisms to reduce the computational costs of computing Shapley Values. The motivation of this paper is well described and the proposed method is technically sound.

**Pros**:
1. This paper is well-motivated and the proposed method is technically sound.
2. The experiments are comprehensive and convincing.

**Cons**:
1. The technical contribution of this paper remains unclear. The authors should clarify the difference between the proposed sampling method and existing methods.
2. Baselines are kind of out of date, more recent baselines should be compared.
3. In table(5), the time consumption of SVXSampler and GNNShap variants are changing randomly with respect to different numbers of nodes and edges, authors should make further analysis of such results.

**Questions:**

See Cons.

**Reviewer Confidence:**

4: The reviewer is certain that the evaluation is correct and very familiar with the relevant literature

**Scope:**

3: The work is somewhat relevant to the Web and to the track, and is of narrow interest to a sub-community

---

### Official Review · Reviewer_QTcW · 2023-11-24

**Novelty:** 3
**Technical Quality:** 4

**Review:**

Strengths:
1. The adaptation of Shapley values from game theory to GNN interpretability is a natural fit, and I am pleased to see the authors develop work in this direction.
2. The authors provide an excellent and clear summary of current Shapley value-based interpretability algorithms, which is beneficial for the development of this field.
3. The architecture design in the method section is well done, very clear and easy to read, friendly to the readers, and the logical structure is also well designed.
4. A wealth of experiments validate that the proposed method has clear advantages over traditional Shapley value-based interpretability algorithms.

Weaknesses of the Paper:
1. There are instances of overclaiming, such as the statement in the contribution section that "GNNShap detects many unimportant edges that can be removed from the graph to expedite GNN inferences." Without substantial experimental validation, this should not be presented as a contribution. To my knowledge, most interpretability papers do not mention this in the contribution section because the impact of directly removing unimportant subgraphs is unpredictable.
2. Section 3.3 needs a clearer motivation for the innovative design of the weights. The current article only explains how the new design of the weights works, as shown in Figure 3. However, after reading the entire Section 3.3, I still cannot fully understand why the previous weights (like the green line in Figure 3) are detrimental to the calculation of Shapley values. I think some theoretical explanations or ablation experiments are needed on this point.
3. A major issue currently facing post-hoc interpretability is the OOD problem, meaning there is a distribution difference between the subgraph and the original graph. This can lead to traditional removal-based evaluation methods (such as fidelity) possibly not truly reflecting the real effect of the interpretability algorithm. Therefore, I think the authors should consider adding datasets that include ground truth and use ground truth-based evaluation metrics (such as recall and precision) to assess the pros and cons of interpretability algorithms.
4. Baselines for experiments are missing. Although I understand that the authors have already demonstrated that the proposed method is better than most known Shapley value-based methods, many of the latest post-hoc interpretability methods are not based on Shapley values, and they are still worth comparing, such as RCExplainer, ReFine, Gem, SubgraphX, etc.

**Questions:**

See the weaknesses

**Reviewer Confidence:**

3: The reviewer is confident but not certain that the evaluation is correct

**Scope:**

4: The work is relevant to the Web and to the track, and is of broad interest to the community

---

### Official Review · Reviewer_PhfE · 2023-11-28

**Novelty:** 4
**Technical Quality:** 5

**Review:**

## **Summary**

This paper proposes a Shapley value based explainer for GNN with better efficiency. The authors first prune the copmutational graph for a node, then apply parallelized fast sampling with better coverage and utilize a efficient matrix multiplication based Shapley value computation to generate explanation. Experimental results show that the proposed method can find explanation with higher fidelity scores in a shorter time.


## **Strong Points**

S1. This paper achieves a better efficiency in Shapley value based GNN explanation.

S2. Experimental results show that it can find explanation with better fidelity scores.


## **Weak Points**

W1. Shapley value is claimed as non-structure-aware in GStarX while Hamiache-Navarro (HN) value used in GStarX is known to be structure-aware. Can the authors elaborate the advantage of using Shapley value instead of HN value?

W2. From reading the paper, it seems that the main contribution is the efficiency improvement compared to EdgeSHAPer, GraphSVX and SubgraphX. Is it really true that the proposed method could handle all different task scenarios for these three methods? It would be better to provide more in-depth discussion about the similarities and differences between this work and the others.

W3. Figure 2 is confusing: (1) Traditionally when we talk about computational graph, edge between 1866 and 1701 (and similarly edge between 1862 and 2582) is not included; the current definition of computational graph seems conflicting with the commonly-used one. (2) This graph considers one undirected edge as two directed edges, this may break an undirected graph to a directed graph and may not be intuitive for real-world undirected graphs. I am curious about the intuition behind this choice. It seems that this does not benefit the computation in section 3.5 because, for an undirected edge, you don't need to count it twice in matrix M.

W4. What is the benefit of the proposed method over MCTS-based sampling in SubgraphX?

W5. The authors should provide more details on why $\phi$ is equivalent to computing Shapley value. Right now there is no detail but one function only.

W6. The linear regression model for matrix inversion is very unclear. Why do the authors choose linear regression over SVD? What is the intution of using linear regression? What is the objective function, input and output?

W7. In Table 5, For some datasets, it is much faster than baseline methods, while for some datasets it only has marginal improvement over baseline method (e.g., SVXSampler on Coauthor-Physics). Can the authors provide more insights about it?

W8. Still in Table 5, it seems that the increase in running time does not have any pattern. The increase from GNNShap 10k to GNNShap 25k to GNNShap 50k is neither linear nor sublinear nor superlinear based on the provided results. Is it possible to analyze the complexity of generating explanation by GNNShap?

**Questions:**

Please see weak points above.

**Ethics Review Description:**

NA.

**Reviewer Confidence:**

4: The reviewer is certain that the evaluation is correct and very familiar with the relevant literature

**Scope:**

3: The work is somewhat relevant to the Web and to the track, and is of narrow interest to a sub-community

---

### Decision · Program_Chairs · 2024-01-22

**Decision:**

Accept

**Comment:**

This paper provides a more efficient way to compute shapely values for GNN explanations. All the reviewers liked the paper and also appreciated the extra experimental results conducted by the authors during the rebuttal paper. While some reviewers also noted that a few claims in the paper may be exaggerated, overall this a paper worthy of acceptance.